# Assessing the Reproductive Ecology of a Rare Mint, *Macbridea alba*, an Endangered Species Act Protected Species

**DOI:** 10.3390/plants12071485

**Published:** 2023-03-28

**Authors:** Sara A. Johnson, Janice Coons, David N. Zaya, Brenda Molano-Flores

**Affiliations:** 1Department of Natural Resources and Environmental Sciences, University of Illinois Urbana-Champaign, 1102 S Goodwin Ave, Urbana, IL 61801, USA; 2Illinois Natural History Survey, 1816 South Oak Street, Champaign, IL 61820, USAmolano1@illinois.edu (B.M.-F.); 3Department of Biological Sciences, Eastern Illinois University, 600 Lincoln Ave, Charleston, IL 61920, USA

**Keywords:** endemic plants, ex situ conservation, Lamiaceae, *Macbridea alba*, rare plants, recruitment, seed ecology, seed predator, vivipary, herbivory

## Abstract

Many rare plant species lack up-to-date research about their reproductive ecology, which challenges effective in situ and ex situ conservation, particularly in the face of ongoing environmental and anthropogenic changes. For protected species, outdated and incomplete information also creates barriers to successful recovery planning and delisting. In this study, we gathered a range of reproductive metrics for the federally threatened and state endangered Florida endemic mint, *Macbridea alba* Chapman (Lamiaceae). We collected data at seven populations within Apalachicola National Forest (Florida, USA) and conducted germination trials to estimate reproductive potential. Additionally, we observed a previously undocumented lepidopteran seed predator for the species and confirmed the occurrence of vivipary. The seed set was low with less than 20% of flowers per inflorescence producing seed across populations; however, germination was high with more than 60% of seeds germinating in five of seven populations. When comparing our results to previous research conducted more than 20 years ago, the results were similar overall (i.e., germination, vivipary); however, new information emerged (i.e., herbivore pressure). As *M. alba* undergoes reassessment as a potential candidate for delisting from the Endangered Species Act (ESA) list, this information is critical for assessing recovery goals and decisions regarding the species’ protected status. For recovery needs related to propagation and reintroduction, these results can inform future seed collection and propagation efforts for the species.

## 1. Introduction

Rare and endemic plants are at an increased risk of decline and extinction [1,2] as many of these species exhibit narrow distributions, specialized ecologies, and are low in abundance [3,4]. These characteristics, exacerbated by anthropogenic and natural causes like fragmentation of natural populations, have led to the decline of plant species globally [5,6]. Conservation biologists emphasize that a barrier to effective conservation and recovery planning for rare species is the lack of data regarding species’ reproductive biology and ecology [7]. This type of data can improve conservation efforts for at-risk species by documenting limits to recruitment that can inform in situ and ex situ safeguarding efforts [7,8,9], particularly regarding reintroduction and habitat management plans [10].

The Florida Panhandle is a region within the northwestern part of the state of Florida within the United States of America. This region includes the 10 counties west of the Apalachicola River. It is considered within the United States to be a biodiversity hotspot and harbors many state and federally listed plant species [11], including *Macbridea alba* Chapman (Lamiaceae, *M. alba*, from hereon). *Macbridea alba* is both federally threatened and state endangered. While the species is considered locally abundant, it is geographically restricted to a range of specific habitats in four counties of the Florida panhandle region, or the northwestern portion of Florida in the United States of America [12]. During the 1990s and early 2000s, several *M. alba* studies provided information about pollinators, population genetics, reproductive output, and germination to assist with its conservation [13,14,15,16,17]. *M. alba* is currently a candidate for delisting from the Endangered Species Protection Act list, yet recovery plans are based on outdated information about the current population size, distribution, and reproductive potential of this species, presenting persistent obstacles to conservation efforts [18].

Much has changed throughout the species’ range in the decades since those earlier studies. With increased pressure on native populations from current and projected environmental change [19], increased stochastic weather events [20], land use change and development [21], threats to pollinator services [22], and the encroachment of exotic and invasive species [23], it is important to provide an updated account of the species’ status. In the case of *M. alba*, few populations are known outside of protected areas and public lands, and those that persist are at a high risk of further decline [12]. In addition, few ex situ collections exist to supplement natural populations. Efforts to propagate, maintain ex situ collections, or reintroduce species to new habitats should be informed by a thorough review of the current life history, habitat specificities, and reproductive ecology. If delisting is the final objective for U.S. rare species, the most up-to-date information regarding their biology and ecology should be a priority to support such delisting.

The goals of this study were to (1) document population size and the number of total flowers per stem among seven populations during one sampling period within Apalachicola National Forest (Tallahassee, FL, USA), (2) document fruit and seed production and germination success for each population, (3) investigate potential variables (i.e., vivipary, herbivory) correlated with *M. alba* seed ecology and possible recruitment, and (4) investigate how reproductive output, germination, and herbivory are correlated to population size and total number of flowers. We collected infructescences from each population to document reproductive output (e.g., flowering, fruit set, and seed set) and conducted germination studies to compare our findings to previous research [14,16]. Based on previous work conducted with the species [14,16], we expected *M. alba* seeds to exhibit high germination and seed viability across populations; however, we expected reproductive output to vary by population. These results will aid in the safeguarding and management of *M. alba* populations in the face of current and future change to advise its protected status.

## 2. Results

### 2.1. Fruit Set, Seed Set, Vivipary, and Herbivory

Overall population size ranged from 67 to 800 individuals with a total number of flowers ranging from 177 to 865 across populations (Table 1). The fruit set was generally low across populations with a maximum documented fruit set of 18% (Table 2). Seed set was also low with a maximum documented seed set of 8% (Table 2). Significant differences were observed in fruit set (χ2 = 31.8, *p* < 0.001, df = 6) and seed set (χ2 = 35.7, *p* < 0.001, df = 6) among populations (Table 2).

As many as 33% of calyces across individuals within a population contained a pre-germinated seed (Table 1 and Table 2). Vivipary (i.e., pre-germinated seed within the calyx) was documented in all populations, and significant differences were found among populations (χ2 = 59.1, *p* < 0.001, df = 6). In the case of the transplanted viviparous seeds, about 40% of the original 135 transplants survived after ~14 months. However, after ~20 months, all plants perished possibly due to pest damage from mealybugs and growing conditions (i.e., sensitivity to fertilizer application and possible drainage issues with soil mixture).

The proportion of fruits showing signs of herbivory varied significantly among populations (χ2 = 53.3, *p* < 0.001, df = 6) with up to 37% of calyces per stem across individuals within a population showing signs of herbivore damage (Table 2). Four of the seven sites surveyed exhibited herbivore damage on stems in at least 50% of individuals, with the most herbivore damage documented in 77% of sampled individuals in one population (Table 2). Micro-lepidoptera specimens, as well as pupal cases, were identified as *Endothenia hebesana* (Walker) (Tortricidae; Verbena Bud Moth).

### 2.2. Germination Trials

The difference in the average germination among populations was significant (F_6,26_ = 3.2, *p* < 0.05), and overall, germination was high with five of seven populations exhibiting successful germination of 60% or more (Table 2). The range across replicates and populations was as low as zero and as high as 100% germination with the highest average at 83% (Table 2). The lowest germination across all populations was an average of 33%. Of the 137 transplants from germination trials, approximately 22% of seedlings survived after ~14 months. However, by the end of the study, all seedlings perished (i.e., ~20 months).

### 2.3. Correlations

Population size ranged from 67 to 800 with total number of flowers ranging from 177 to 865 across populations (Table 1). Reproductive output (i.e., fruit set, and seed set), germination, and herbivory were not correlated to population size (all r values < 0.676 and all *p* values > 0.096) or total number of flowers (all r values < -0.236 and all *p* values > 0.610).

## 3. Discussion

For most federally listed species, few will have long term demographic, reproductive, and/or ecological data available to inform conservation strategy. For many other rare species, decades may pass between surveys. This study aimed to document reproductive metrics for *Macbridea alba*, a rare Florida mint, after more than two decades since previously published work. Throughout this time, populations numbers have remained stable; however, the number of extant populations have become primarily restricted to public lands, specifically within Apalachicola National Forest [12]. The conversion of natural habitat to cattle pasture or improperly managed timberlands has enabled fragmentation, fire suppression, and woody encroachment: all factors that could impact long-term survival and recruitment of *M. alba* populations [12,13]. Across populations sampled in our study, fruit set and seed set were low despite high floral output. Germination success varied across populations but was high overall. The inspection of seeds confirmed and quantified the occurrence of vivipary and documented potential threats to *M. alba* recruitment and survival, including the presence of a natural seed predator prevalent across *M. alba* populations. These results show that populations can display significant variation in reproductive output, which has important implications for collection and ex situ propagation efforts. Importantly, our research both supports and adds to the existing body of work on *M. alba*’s reproductive ecology. By providing up-to-date data concerning *M. alba* reproductive biology, our results can help to prioritize recovery and safeguarding efforts for the species, as well as informing the species’ current protected status. Beyond the immediate applications for *M. alba*, this approach is useful when seeking new data to inform conservation efforts, status assessments, and potential delisting of other rare and listed species.

### 3.1. Fruit Set, Seed Set, and Herbivory

The observed fruit and seed set were low across all surveyed *M. alba* populations. Although our study used different metrics from a previous work by Madsen (1999, [14]), we observed similar numbers in reproductive output for the species. Madsen referred to individual clumps and reported seeds per flower while we estimated seeds per flower by remnant calyces by stem per individual. In our study, we documented an average range of 0.1 to 0.3 seeds per calyx across populations compared to an average of 0.45 to 1.49 seeds per flower across populations in Madsen’s study (1999, [14]). In addition, as in the case of Madsen (1999, [14]), higher floral production did not necessarily equate to an increase in fruit or seed set, and various other sources of variation at the population level likely determine ovule success. It is documented that *M. alba* population numbers and floral production commonly vary year to year depending on environmental conditions and burn history [24]. Studies that monitor floral and seed production over time will improve our understanding of patterns in variations of seed production.

Regardless of population size or number of flowers available, there were few developed fruits and seeds observed across populations, and an increase was not correlated with an increase in reproductive output. While there are many potential explanations, including temporal variability, the low reproductive output could in some part be explained by a low occurrence of pollinator visitations. Pitts-Singer et al. (2002, [15]) noted that although bumblebees and other bees were visiting *M. alba*, visitation rates were low. During site visits in 2019–2020, pollinators and their visits were rarely observed (Sara Johnson and Brenda Molano-Flores, personal observations). A recent review by Sheehan and Klepzig (2022, and citations therein, [25]) highlighted the resilience of the bee communities in the longleaf pine ecosystem and the benefits of habitat management for pollinators. With 73% of plants in the longleaf pine savanna relying on insects for pollination [26], additional observations are needed to better quantify pollinator visitation rates and reproductive output within the context of habitat management for *M. alba* and other rare plants. Depending on the species, different patterns could be observed throughout longleaf pine ecosystems (e.g., *Pinguicula ionantha*, [27]).

Another potential explanation for the low fruit and seed set is the prevalence of a newly documented seed herbivore for the species. *Endothenia hebesana* (Walker) is a polyphagous micro-lepidoptera species that feeds on the developing seeds of host plants [28]. This species has been documented to feed on other genera in the mint family, such as *Scutellaria*, *Veronica*, and *Physostegia*, but has not yet been documented on *M. alba* (James Hayden, Florida Museum of Natural History, personal communication). While this naturally occurring herbivore is unlikely the sole cause of low seed set in *M. alba* populations, herbivory was abundant and present in over 15% (ranging up to 77%) of stems across all sampled populations (Table 2). It is unknown whether *M. alba* individuals compensate for increased herbivory during the flowering season [29,30]. Continued pre-dispersal seed predation in perennials like *M. alba* may contribute to overall mortality, reduced recruitment, and limited population growth [31,32,33]. Resource limitation (i.e., lack of light or moisture) exacerbated by competition and encroachment may also contribute to the low observed seed set within *M. alba* populations, as encroachment is a major issue and important focus of habitat management in this region [12].

### 3.2. Vivipary

Vivipary has been documented previously in *M. alba* individuals with around 20% of collected seeds germinating in the calyx [16]. In our study, we found populations with higher or lower levels of vivipary than previously reported (Table 1). In addition, pre-germinated seed accounted for ~25% of all seeds collected across populations. It has been suggested that vivipary acts as a reproductive strategy allowing seedlings to overcome limiting growing conditions [34,35]. While increases in humidity and moisture [34], and references therein may lead to an increase in vivipary, in this study, we did not measure these environmental variables and as such, we cannot say whether these variables play the same role in vivipary for *M. alba*. However, *M. alba* calyces are positioned with an open cup shape at the top of the infructescence, creating a location for water to collect during rainy or humid weather. Based on our data, it is uncertain whether this vivipary is adaptive or incidental, and if the adaptive potential could be context dependent based on local conditions. Future research to document the consequence of pre-germinated seed will help provide clues to the success of vivipary for *M. alba.* For example, do seeds successfully fall to the ground and establish or die due to desiccation.

### 3.3. Germination

The ex situ germination success for *M. alba* was over 50% in five of seven populations. While germination results in Schulze et al. (2002, [16]) for *M. alba* focus on varying treatments of age, stratification, and incubation techniques, overall, final average percent germination across the study ranged from 67 to 85%. Mean percent germination was similar in this study at the maximum range (83%) when compared to Schulze et al. (2002, [16]). Two *M. alba* populations had germination ranging from 33–43%, on the lower range of germinability.

Additional work by Schulze et al. (2002, [16]) highlights the temporal nature of *M. alba* seed viability and the lack of persistence in the seed bank. *Macbridea alba* appears tolerant of seed burial only to a depth of less than five cm, as deep burial likely inhibits the emergence of cotyledons from the soil surface (i.e., germination may occur; however, seedlings do not emerge at the soil level and perish due to lack of light, which may be important for *M. alba* germination) [16]. Additionally, fire suppression, competition by invasive species, or woody encroachment could account for low observed seedlings in some populations [12]. These factors, combined with knowledge of *M. alba* seed ecology, suggest that seedling emergence and survivorship could be a limiting recruitment issue in wild populations, not seed germination. Seedlings are infrequently documented in the field, and it is possible that *M. alba* seedlings require a select set of temporal and habitat conditions to germinate and survive in the wild. As noted by other studies (e.g., [36]), these combined limitations may present a narrow window of opportunity for successful sexual reproduction in this species.

### 3.4. Future Work

As with any listed rare plant species, additional work is needed to facilitate delisting. For example, due to the increased pressure from frequent stochastic weather events, isolation of natural populations, and shrinkage of the *M. alba* natural range outside of protected areas, collection and protection should strive to maintain and enhance genetic diversity where possible in both in situ and ex situ conservation efforts. Previous research implicates inbreeding depression as a potential risk to successful *M. alba* recruitment [17]; however, additional research is required to understand the current distribution of genetic diversity across populations and to reassess if inbreeding depression is still a concern. In addition, unknown implications of low outcrossing and limited genetic variability caused by the fragmentation of populations may leave existing *M. alba* populations vulnerable to continued habitat and climate change.

*Macbridea alba* also has been documented to spread frequently by rhizome via asexual reproduction, and clonal establishment may vary among years or sites. Further work is necessary to specify the primary reproductive strategy of *M. alba* and the frequency of sexual and asexual reproduction, as well as the environmental drivers related to the prevalence of reproductive strategy. For example, how does asexual reproduction change how we define relatedness within populations, and what is the role of vivipary as part of the reproductive strategy of this species? In addition, understanding the primary form of reproduction may help to explain the infrequency of seedlings encountered throughout populations [12,14]. It is possible that *M. alba*’s tendency towards sexual or asexual reproduction may fluctuate due to habitat condition and may vary at different times or seasons [37,38,39]. Developing a better understanding of these reproductive strategies in conjunction with current population genetics, habitat conditions, and frequency of fires will provide insights into the most productive strategy for maintaining diverse populations at in situ or ex situ locations. Having these data in combination with other datasets, such as long-term monitoring data, could facilitate the development of population viability analyses [40,41] or matrix projection models [42] to inform whether *M. alba* populations are stable, increasing, or declining overall. In addition, demographic models can help link abiotic and biotic factors in the environment to vital rates and overall fluctuations in abundance and reproductive effort [43].

## 4. Materials and Methods

### 4.1. Study Species

*Macbridea alba* Chapman (Lamiaceae, white birds-in-a-nest) is a federally threatened and state endangered perennial herbaceous mint restricted to Bay, Franklin, Gulf, and Liberty counties in the Florida panhandle region, or northwestern portion of the state of Florida in the United States of America [44]. The species was listed as threatened under the ESA in 1992 as threats of habitat degradation caused by poor management practices for timber and cattle were increasing [12]. This fire-adapted and disturbance-dependent species is monitored and managed by state and federal agencies throughout public lands where it persists. Populations are associated with grassy pine flatwoods of the longleaf pine (*Pinus palustris*) ecosystem, but individuals are commonly observed across a range of conditions from wet savannas and sand hills to disturbed roadsides [12]. Individuals typically produce one or more, often branched stems up to ~45 cm in height and are conspicuous when in bloom from May through July. Flowers are bisexual with bright white corollas arranged in a terminal inflorescence (Figure 1), and seeds mature from July to September. Seeds likely have low germination rates in the field and recruitment (i.e., and seedlings) has not been recorded in the wild [12,16].

*Macbridea alba* reproduces via rhizome and by seed, producing up to four nutlets per flower. Vivipary occurs occasionally with seeds germinating within the calyx [16]. In addition, *M. alba* is self-compatible and pollinated by bumblebees [15]. Genetic research shows about 92% of genetic diversity is found within populations [17], and genetic diversity may be lower than other perennial Florida mint species [45]. Drought or extreme weather may reduce reproductive output or result in temporary dormancy until conditions improve [15,17]. Research suggests that *M. alba* may be a poor competitor with other plants, as it may require bare ground to germinate and could be restricted by its inability to tolerate shade [46]. Lastly, stored and buried seeds remain viable for up to six months; however, viability rapidly declines after one year. The absence of a persistent seed bank and lack of innate dormancy create a narrow temporal recruitment window for the species [16].

### 4.2. Study Area

Apalachicola National Forest (ANF) is home to over two-thirds of occurrence records for *M. alba*, as well as multiple long-term monitoring plots maintained by the Florida Natural Areas Inventory (FNAI) [47,48]. Fire suppression and habitat modification has fragmented the once extensive longleaf pine ecosystem of the coastal plain, but intact habitat persists within the protection of ANF and surrounding public lands [12]. Exacerbated by fire suppression and poor forest management practices on both private and public lands, encroachment by woody species has introduced competition, which challenges the survival of *M. alba* and associate herbaceous species within this fire prone region [12,24]. Furthermore, this area is managed by state and federal entities with mechanical and chemical removal of woody species, mowing, and frequent burning; however, burn frequency varies across compartments within ANF.

Preliminary surveys were conducted at a selection of previously reported records (n = 98) to estimate population size. Due to the quantity of sites surveyed during the study period, some population sizes were estimated, and some represent exact counts. Seven populations (at least one kilometer apart) were selected in ANF for seed collection based on their approximate population size and varying habitat and management conditions. Habitat condition ranged across populations in terms of the level of woody encroachment, the cover of vegetation at the canopy, understory, and ground levels, and the microtopography of the site from upland to wetland habitat. Fire compartment data for ANF [49] was utilized to determine the burn history (e.g., time in years since the last burn) for each population.

### 4.3. Seed Collection

A range of approximately 20 to 98 *M. alba* individuals were haphazardly selected for seed collection from each population based on estimated abundance. In July 2019, after flowering but before fruit development and seed dispersal, individuals were identified by tracing the stem to the base or basal rosette of each plant. One flowering stem per rosette was bagged with a mesh bag. Because a flowering stem often exhibits a branching inflorescence, all flower heads within that flowering stem were bagged. There is a low chance of shading from mesh bags to the infructescence of each plant. In September of the same year, stems were clipped below each bagged infructescence and were brought to the lab for dissection. The total infructescences collected (measured by infructescences per mesh bag) were counted and the total number of calyces (which also estimates flower production) per infructescence were removed and counted. Calyces that contained at least one seed were counted as a fruit. Calyces were dissected to expose seeds, which were then removed, counted, and pooled at the population level. Fully developed seeds were plump and a light tan, whereas any appearing soft or dark in color were considered dead or undeveloped and were discarded.

### 4.4. Reproductive and Population Metrics

For each individual (i.e., stem), the number of infructescences per stem, the total number of calyces (to estimate floral output per stem), and the number of fruits (i.e., total number of calyces containing at least one seed) were recorded. Fruit set was estimated by counting the number of fruits as a percent of the total number of calyces produced per stem:*% fruit set = # fruits/# calyces per stem × 100*

In addition to developed seeds, viviparous seeds were encountered during seed extraction, and therefore, fully developed, and viviparous seeds were combined to calculate total set per stem. Each *M. alba* flower produces one ovary with a gynobasic style that may produce a fruit with up to four nutlets, as each of the four ovary lobes may produce a seed/nutlet. Potential seed set per stem was determined by multiplying the total number of calyces per stem by four. Therefore, seed set was defined as a percent of total set from potential seed set.
*total set = developed seed + viviparous seed*
*% seed set = total set/potential seed set × 100*

The percentage of calyces with and without viviparous seed was documented, as well as the total percentage of viviparous seeds of total set per population. All viviparous seedlings gathered during collection were transplanted into 5 × 5 × 5 cm pots in a potting mixture consisting of 3-parts potting soil (peat and perlite), 3-parts sand, 2-parts perlite, 1-part lava rock, 1-part horticultural grit, and ½-part white pine (*Pinus strobus*) needles (collected from the researcher’s neighborhood) and ½ part fine orchid bark (Dalton’s Orchiata, Matamata, New Zealand), in ratio of volume. Transplants were raised in a temperature-controlled (see *Germination Trials*) greenhouse from September 2019 onward, and survivorship was documented.

Herbivore damage was documented in several *M. alba* individuals, and it was considered to be herbivore damage if there were holes or damage present on the calyces of the infructescence. The number of calyces with damage per stem was counted during dissection. The number of stems with at least one incidence of herbivory was also summarized for each population. Upon inspection of calyces, insect herbivores (i.e., adults and pupal cases) were collected.

### 4.5. Germination Trials

Greenhouse germination trials were conducted in the fall of 2019 for a period of two months at the University of Illinois at Urbana-Champaign Plant Sciences Lab Greenhouse. Germination trials were conducted within 2 months of collection, as work by Schulze et al. (2002, [16]) documented successful germination for seeds up to 6 months in age and a lack of dormancy for the species. For germination trials, the number of replicates and the number of seeds per replicate varied by population based on pooled seed collection totals. Seeds were placed in 100 mm by 15 mm plastic Petri dishes lined with one sheet of 90 mm diameter Whatman™ grade 1 filter paper before adding 2 mL of distilled water (dH20) to each. Petri dish edges were sealed with Parafilm™ to reduce evaporation and dishes were placed on a bench in a controlled greenhouse set to a 14 h light (7:15 a.m. to 9:15 p.m.) and 10 h dark period with day temperature set to 22/25 °C and a night temperature of 8/12 °C. Every day or every other day when seeds were checked for germination, position was randomized to avoid a “block effect”. Germination was considered as the emergence of the radicle. Few germinants were observed by day six of the trial, and an additional two pieces of filter paper were added to help retain moisture and prevent seeds from drying out. At this time, Captan™ (an antifungal agent, 50% Wettable Powder, BONIDE Products LLC, Oriskany, NY, USA) was sprinkled over the filter paper and seeds to inhibit mold growth. Additional dH20 was added as needed and petri dishes were resealed with Parafilm™ each time. Sprouted seeds were immediately removed and transplanted into 5 × 5 × 5 cm pots in a soil mixture as outlined above. Transplants were raised in greenhouse conditions described above and survivorship was documented. Transplants were watered as needed (approximately every other day or so), and transplants were fertilized monthly with MSU orchid fertilizer (19-4-23, N-P-K, Michigan State University (MSU) formula, East Lansing, MI, USA). Pesticides were used at least twice during this time by greenhouse staff to control greenhouse pests, such as mealybug and thrips.

### 4.6. Statistical Analysis

To examine differences among populations in fruit set, seed set, amount of viviparous seed, and the number of calyces with herbivore damage, Kruskal–Wallis tests followed by Dunn post hoc tests were conducted. A one-way ANOVA followed by a Tukey post hoc test was used to test differences among populations regarding the overall germination percentage. To standardize the presented data, the average proportions ± SEs per stem for each population are reported, and significance was determined at alpha = 0.05. To determine if reproductive output, germination, and herbivory are correlated to population size and total number of flowers, Pearson Product Moment and Spearman Rank Order correlations were conducted. All analyses were performed using R Version 3.6.3 [50]. The following packages were used: *agricolae* [51], *car* [52], *dplyr* [53], and *rcompanion* [54].

## 5. Conclusions

This study provides up-to-date data for an endemic mint, *Macbridea alba*, that has remained stagnant on the Endangered Species Act (ESA) list since its listing. Research conducted over 20 years ago provided important baseline data for the species; however, conditions have changed significantly in the species’ native range throughout that time. For rare plant species, particularly ESA protected species, updated data can provide useful information for evaluating the effectiveness of current conservation and management plans by documenting potential changes in biology, reproductive output, and recruitment. The information in this study contributes to the outlined recovery needs for the species, particularly the recovery goal of improving in situ and ex situ propagation and reintroduction efforts. With updated data, we hope that conservation practitioners can prioritize recovery goals to protect populations where they persist, and if possible, delist species.

## Figures and Tables

**Figure 1 plants-12-01485-f001:**
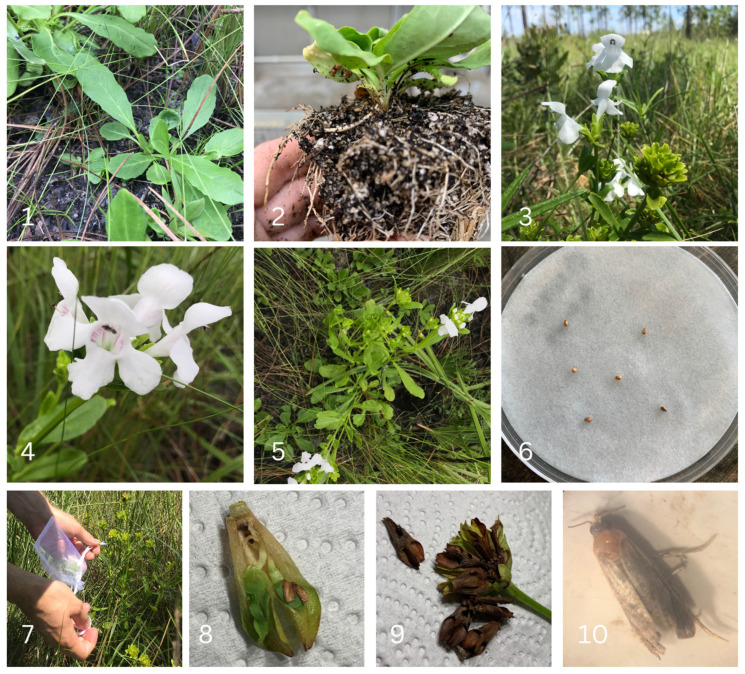
Photos of plant and reproductive parts of *Macbridea alba* individuals with one or multiple stems per rosette. Photos from left to right: (**1**) basal rosettes in situ; (**2**) basal rosette and root system of ex situ grown individual; (**3**) multi-stemmed inflorescences of single individual (side-profile); (**4**) single inflorescence; (**5**) multi-stemmed inflorescences of multiple individuals (top-profile), (**6**) *Macbridea alba* seeds in petri dish for germination trial; (**7**) bagging single stem of multi-stemmed individual in the field; (**8**) multiple pre-germinated seeds within single calyx; (**9**) evidence of herbivory on calyces of *Macbridea alba* infructescence; (**10**) specimen of *Endothenia hebesana* (Walker) or the Verbena Bud Moth discovered in specimen bag during collection.

**Table 1 plants-12-01485-t001:** Summary of the number of sampled individual stems, estimated population size, total flowers per population, and estimated average flowers per stem. Average number of seeds per calyx, number of developed seeds, and viviparous seeds collected per population. Percent viviparous seed is a percent of total seed set (developed seed + viviparous seed) for each population. Percent calyces per stem with at least one incidence of vivipary across sampled individuals per population.

Population n = Indv	Population Size	Total Flowers (Avg/Stem) *	Avg Seeds per Calyx	Developed Seed	Viviparous Seed	% Viviparous Seed	% Calyces with Vivipary ** ±se
1 n = 49	800	570 (12)	0.3	78	80	51	32.7 ± 5.9 ^a^
2 n = 48	102	332 (7)	0.1	15	3	17	2.5 ± 2.1 ^b^
3 n = 19	170	177 (9)	0.3	36	18	33	21.5 ± 8.6 ^ab^
4 n = 52	250	546 (11)	0.1	33	7	18	2.9 ± 2.0 ^b^
5 n = 48	250	420 (9)	0.1	40	3	7	3.2 ± 2.2 ^b^
6 n = 98	445	865 (9)	0.2	164	20	11	4.8 ± 1.3 ^b^
7 n = 48	67	452 (9)	0.1	39	4	10	4.4 ± 2.9 ^b^

* Flowers are estimated based on number of calyces. ** Letter denoting significant differences are based on the KW-Dunn’s test results for medians.

**Table 2 plants-12-01485-t002:** Summary of the number of sampled individuals and the average fruit set and seed set (including developed seed and viviparous seed) per stem for each population. Percent calyces per stem and percent of stems per population with at least one incidence of herbivory. The average percent germination for each population as well the range of germination across replicates within a population.

Population n = Indv	% Fruit Set * ± se	% Seed Set * ± se	% Calyces with Herbivore Damage * ± se	% Stems with Herbivore Damage	% Germination ** ± se (Range)
1 n = 49	14 ± 3 ^a^	7 ± 2 ^a^	7.8 ± 3.8 ^a^	73.5	62.6 ± 9.2 ^ab^ (33.0–87.0)
2 n = 48	4 ± 1 ^b^	1 ± 0 ^b^	21.2 ± 3.7 ^ab^	54.2	33.3 ± 17.6 ^b^ (0.0–60.0)
3 n = 19	18 ± 5 ^ac^	8 ± 2 ^ac^	5.5 ± 3.2 ^b^	15.8	83.4 ± 7.4 ^a^ (67.0–100.0)
4 n = 52	3 ± 1 ^bc^	1 ± 1 ^bc^	13.7 ± 3.3 ^b^	32.7	43.4 ± 8.5 ^b^ (17.0–67.0)
5 n = 48	7 ± 3 ^bc^	3 ± 1 ^bc^	36.5 ± 4.3 ^a^	77.1	63.2 ± 9.7 ^ab^ (33.0–83.0)
6 n = 98	9 ± 2 ^abc^	5 ± 1 ^abc^	33.3 ± 3.2 ^a^	68.4	72.0 ± 5.6 ^ab^ (50.0–80.0)
7 n = 48	5 ± 2 ^bc^	2 ± 1 ^bc^	9.9 ± 2.0 ^b^	22.9	60.0 ± 6.6 ^ab^ (50.0–83.0)

* Letter denoting significant differences are based on the KW-Dunn’s test results for medians. ** Letter denoting significant differences are based on the ANOVA-Tukey test results for means.

## Data Availability

Due to the nature of species listing status, raw and/or locality data can only be made available through the permission of the U.S. Fish and Wildlife Service. For other questions, please contact the corresponding author.

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
