# Peer review of "Assessing the Reproductive Ecology of a Rare Mint, Macbridea alba, an Endangered Species Act Protected Species"

_plants, 2023, doi:10.3390/plants12071485_

Round 1
Reviewer 1 Report (New Reviewer)
Review of “Assessing the reproductive ecology of a rare mint, Macbridea alba, an endanger species act protected species."
General comments
This is a study of flower, fruit, and seed production, as well as seed germination, and seedling survival of several populations of M, alba, a mint endemic to 4 counties of the Florida panhandle. It is a follow up study from studies that were conducted 20 years earlier. The findings of this study are mostly consistent with the findings of the earlier studies. This study found herbivory that was not included in the previous studies.
The paper is well written. It is clear and easy to understand. The introduction and discussion relate this study to other studies of conservation biology. The data are presented in a clear and straightforward manner. I find only minor points for suggested changes.
This paper adds incremental knowledge of the reproductive activity of this species. It does not add significantly to our understanding of this species. The germination and seedling survival analyses were conducted in the lab and greenhouse. It would have been much more informative, but also much more involved if these experiments had been conducted in the native populations. Most plants do not have vivipary, it would have been interesting to understand the different in-situ survivorship of seedlings developed by vivipary compared to seedlings developed from seed that germinated on the ground. Since the paper discusses possible reasons for potential population decline, it would have been especially helpful to study germination and seedling survival in the variety of habitats in which the species grows. The text suggests changes in the vegetation of the native populations that could negatively affect these populations but does not test them.
Specific comments
Introduction, Line 72: This sentence should use past tense.
Materials and Methods, section 4.1: More description of the plant would be helpful. For example, there is no mention that this plant is a perennial. Also, it would be helpful to know when seeds germinate in natural populations.
Materials and Methods, line 327: Comment on the shading effect of bagging on the seeds.
Materials and Methods, line 354: Is “parts” for the potting mix in volume or mass?
Author Response
We appreciate the comments of the reviewer as they have greatly improved the manuscript.
See below responses to the specific comments:
Introduction, Line 72: This sentence should use past tense. Corrected
Materials and Methods, section 4.1: More description of the plant would be helpful. For example, there is no mention that this plant is a perennial. Also, it would be helpful to know when seeds germinate in natural populations. The plant is listed as perennial in line 269. Information added to lines 279-281.
Materials and Methods, line 327: Comment on the shading effect of bagging on the seeds. Information added to lines 327-328.
Materials and Methods, line 354: Is “parts” for the potting mix in volume or mass? Parts refers to ratio of volume.
Reviewer 2 Report (Previous Reviewer 2)
The authors of the paper introduced many changes that greatly improved the quality of the presented research. I am convinced that such documenting studies on rare species are very valuable and important. In my opinion, the manuscript in this form is suitable for publication. I found only 2 places to change text editing: p. 2: 77%, p. 5: .. seasons [..].
Author Response
We appreciate the reviewers' comments as they have greatly improved the manuscript.
Please see the responses to specific comments below:
I found only 2 places to change text editing: p. 2: 77%, p. 5: .. seasons [..]. I found 77% referred to on page 4 and have removed the space between number and %. I have removed the . after seasons on pg 6.
Reviewer 3 Report (New Reviewer)
This study examines an endangered species to update the status of the populations and document variation among them, for the purpose of re-evaluation of best management practices. The researchers note interesting features of its reproduction (seeds germinating in fruit!) and threats to its fitness (herbivores and their damage) that were previously unknown for the species. The study was carefully conducted, clearly described, beautifully illustrated, and the data analyzed appropriately. It will be a valuable addition to the literature on rare plant conservation. I have made some small edits on the presentation/wording of sentences in parts of the manuscript, making those changes and comments using the comments tool in Adobe pdf, file attached. I have two more general comments that I will list here:
Figure 1 is beautiful, very nice photos that illustrate important features of the study. It would be nice to also see a picture of the viviparous seeds in a calyx, and maybe some herbivore damage and a caterpillar or moth herbivore.
In the Lit. Cited, be sure to put latin names in italics and the publication year should be in BOLD

Author Response
We appreciate the reviewers comments as they have greatly improved the manuscript. We have addressed all in-text comments as well as specific comments listed below:
Figure 1 is beautiful, very nice photos that illustrate important features of the study. It would be nice to also see a picture of the viviparous seeds in a calyx, and maybe some herbivore damage and a caterpillar or moth herbivore. I have updated figure 1 to include some of these features.
In the Lit. Cited, be sure to put latin names in italics and the publication year should be in BOLD. I have gone through the literature cited and bolded the publication years and reviewed the latin names for italics.
This manuscript is a resubmission of an earlier submission. The following is a list of the peer review reports and author responses from that submission.
Round 1
Reviewer 1 Report
I think that studies of the biology of rare plant species can make important contributions to the conservation of such species. However, the current study is not a good example. Population size and reproduction of a rare plant were studied in only seven populations and it was then attempted to relate population variables to environmental conditions, which were, however, estimated very roughly. Results based on such a datasets are always problematic. correlations between all variables are calculated in the hope of finding some "significant" relationships without any clear hypotheses and without adjusting p-values for the inflated risk of error.
Moreover, insufficient information is given on the methods of analysis and how certain statistics were calculated. Nevertheless, it seems clear that several of the analyses are flawed. The manuscript is poorly written, often imprecise and sentences are not logical. Tables and figures are of poor quality and contain errors.
The manuscript could be shortened by 50% without loss of information. For example, a reader will not be interested in the results for each of the seven study populations separately. There are lots of general statements and even whole paragraphs which are true, but trivial and not based on the results of the current study.
I think the title is misleading. It is far too general for a study of a single species. Moreover, if the "importance of revisiting" is stressed, I would expect a comparison of current results with earlier results in the results part showing strong differences between earlier and new results. Instead there is a bit of that in the discussion, but it is rather inconclusive and the study does not really provide much that is new. It remains also remains unclear what the results of the study have to do with delisting.
It is claimed that the study highlights the value in monitoring and revisiting rare plant species over time to document potential changes in demography, viability, and recruitment as environmental conditions and management change. However, I do not see how the current study highlights the value of monitoring. No changes in demography, viability or recruitment were studied.
Specific remarks
Abstract:
- What temporal "trends" were studied?
- What "patterns held"? No patterns in germination or vivipary were studied, it was simply observed that the range of observed values was not very different from what had been observed earlier, which is hardly surprising.
Introduction line 2: "often"? Rare species are by definition low in abundance.
line 4: "Combined?" - narrow distributions etc. do not result in the decline of species, it is the anthropogenic factors.
line 4: "disturbance". Fragmentation is not due to disturbance.
line 6: Why "potential" limits of growth? I would suggest that the actual factors limiting population size are of interest.
line 7: To what refers "where"?
page 2, line 1: unknowns => lack of knowledge
page 2, line 5: Why is "persistence" important?
page 2, line 6: "it is important to assess potential change in species status in the context of the
current climate and environment" - I would then expect in the paper an analysis of changes in population size in relation to changes in climate or similar. This is not the case.
page 2, line 18: "during"?
page 2, line 2: I do not find much about "trends worthy of future research". The respective paragraph (page 7) is extremely general and could have been written without the current study, even without knowing much about the study species. It probably applies to all species of the studied habitat.
Table 1: Apart from people working with the studied species in the studied area, no reader will be interested in the data for all 7 populations. Present aggregated values, perhaps with ranges in addition.
- How was population size determined or rather estimated? Some numbers ("800", "250") are obviously rough estimates, whereas others ("445") look like actual counts.
- Text to the table: Describing in which of the sites fruits, vivipary etc. were highest is not of interest to the reader, because a reader does not know the sites.
- "Total set" is superfluous as it is simply the sum of "total seed" and "Total viviparous seed" (by the way, not named very appropriately). However, there is something wrong. "Total seed" and "Total viviparous seed" refer according to the text to the numbers collected per site. However, "Total set" which is the sum of those two values is supposed to represent values per stem (see text). This does not make sense.
- How were the standard errors per population for fruit set and overall seed set calculated? What was the unit of replication? I guess that you calculated proportions per stem but this should be described clearly. It would then also be important to point out that the values are mean proportions per stem and population, not proportions per population.
- Posthoc comparisons among populations are not of interest, because the reader does not know anything about the populations. These populations are not treatments. Thus, that e.g. population 3 was different from population 1 is of no interest.
Table 2:
"% Calyces w/ Vivipary" ± se and "% Calyces w/ Herbivore Damage ± se"
- What is "w/" ?
- How were the standard errors per population calculated? What was the unit of replication? I guess that you calculated proportions per stem but this should be described clearly. It would then also be important to point out that the values are mean proportions per stem and population, not proportions per population.
- Posthoc comparisons among populations are not of interest, because the reader does not know anything about the populations. These populations are not treatments. Thus, that population 3 was different from population 1 is of no interest.
- According to the methods seeds were pooled per population. How were the mean proportions calculated? How was the standard error calculated? What was the unit of replication for the germination proportion? Explain in the methods.
- "Means are reported" - The KW-test used for vivipary and herbivory compares medians, not means. Thus, the values presented and the statistical analyses do not match.
page 3, line 4: I do not understand the number of 135 plants. The number of viviparous seedlings was much higher and I understood that all were grown.
page 3, line 8: "Herbivory was also significant (ê“2 = 53.3, P < 0.001, df = 6)" - What is this supposed to mean? I guess you mean "The proportion of fruits showing signs of herbivory varied significantly among the studied populations"
page 4 line 2: "Percent germination ranged from zero to 100 across replicates within sites" - From the df given, it follows that there were 32 replicates, what were the replicates?
Table 3. I find this a highly dubious exercise. Each of the variables was correlated with all others without any clear hypotheses, resulting in 44 correlations. The finding that overall there were more "significant" relationships than expected by chance does not increase the reliability of the individual correlations. You should at least adjust p-values by a false discovery correction.
Moreover, the absence of significant relationships between certain variables does not mean anything as there were only 7 replicates and the statistical power was thus very low.
- Why state in the table header (<0.05) if exact p-values are given?
page 4. "fruit set and seed set decline, and drop significantly when herbivore damage is present" - How was the significance of the relationships determined? The two relationships in Fig. S2 and S3 do not look very strong.
Figure S1 and S2.
- Do not capitalize all first letters.
- How were the correlation coefficients calculated? What were the replicates?
- The axes legends do not make sense. Percent calyces ranging from 0 to 1? Either rename as "Proportion of calyces" or multiply scale by 100.
- According to the methods, mixed model analyses with binomial variables were carried out, although this is not described very clearly. Regression fits then apply to logit-transformed data. However, the scales of the figures are linear. Relationships for back-transformed data should be curves, not lines.
Table S3. I do not understand the table. What is it supposed to show? Comparing AIC-values for analyses with different dependent variables does not make sense. Moreover, I guess seeds set values are per fruit, while fruit set values must be for stem, although this is not explained.
Akaike weights for analyses with one explanatory variable are trivial. Why are there two Null models below each other in the table?
Figure S1. Regression lines do not make sense if rank correlations are calculated.
Page 5: "Beyond the immediate applications for M. alba, this approach
to revisiting plant reproductive ecology could provide new data critical to improving the
conservation, reintroduction efforts, and potential delisting of other rare and listed species." - I do not see evidence for a new approach that justifies such a statement.
page 5, lines 17-18: "compared to the potential" - % fruit set and seed set is by definition relative to the potential values.
page 5, lines 20: To what "trends" are you referring?
page 5, 25: fruit and seed set can of course be independent of the number of flowers, this is not surprising
page 5, 27: " It is documented that M. alba population numbers and floral production commonly vary year to year, depending on environmental conditions." - This was not shown in the current study. If the statement is based on another study, that one should be cited.
page 5, 27: "in the discrepancy"?
"It is possible that habitat changes associated with our sites could explain the observed low visitation rate" - Low in comparison to what? You did not study changes in habitat conditions.
page 5, line 42-45: There are many statements such as this one, which are very general and true, but have little to do with the results of the current study.
page 5, line 52: It is stated that Endothenia feeds on the developing seeds of host plants. It is then hardly surprising, that no seed predation was observed once fruits were fully matures
page 6, line 2: "cause of seed limitation in M. alba populations". - It is nowhere shown that populations of M. alba are seed limited. Seed limitation requires that an increase in seed production would not result in an increase in the population growth rate.
page 6, line 8: Vivipary. Not enough information is given to allow a reader to interpret levels of vivipary. What happens with seeds that have germinated in the fruits? Will the seedlings fall to the ground and grow or will they die due to desiccation?
page 6 germination: " Ex-situ germination success for M. alba was over 50% in five of seven populations, despite low seed set". - Why "despite"? Germination is often completely independent of seed set.
"reflecting that an increase in propagules can potentially improve the chance of germination" - What possible mechanism could be responsible for such an effect?
" whereas deeper burial inhibits the emergence of cotyledons from the soil surface, thus triggering germination" - Why "thus"? Does not make sense to me.
The last paragraph on germination is speculative and not based at all on own results.
page 7. "In addition, population size was negatively correlated with increased
time since burn, reflecting previously documented trends that M. alba reproductive output may diminish with increased time since burn". - Flawed logic. Population size of a perennial plant like M. alba is not likely to be closely related to reproductive output.
"This study highlights the value in monitoring and revisiting rare plant species over time to document potential changes in demography, viability, and recruitment as environmental conditions and management change." - I do not see how the current study highlights the value of monitoring. No changes in demography, viability or recruitment were studied.
Reviewer 2 Report
The manuscript entitled: “The Importance of Revisiting the Reproductive Ecology of Rare Plants for Delisting Purposes” concerns the reproductive biology of one of the endemic plant species in Florida. The study brings new data regarding the size of selected populations, seed production, germination ratio, and predation ultimately affecting seed rain. Such studies are essential for conservation efforts at various levels, including administrative ones. Undoubtedly, the basis for further action should be an inventory of currently existing populations and an estimate of their size. In view of possible decisions to delist the species from the federal list, it is undoubtedly necessary to publish the results.
The study was very well planned and executed. Also, the work presented is prepared in a perfect way. The following remarks may help the authors to improve/clarify the manuscript.
Remarks:
1. Title: Please consider a change in the title: The Importance of Revisiting the Reproductive Ecology of Rare Plants for Conservation Purposes
2. Results, table 1, & 4. Study species:
As the species is perennial, usually produces one stem but it could reproduce via rhizome please clarify (and use consequently) in the text and table what was the measure: individual or stem. Please put a better description of the species in section 4 by adding the information of species form i.e ..This is a perennial herb producing usually one stem measuring usually 30-40 cm tall (but as you wrote 1.5 m in your sites)
3. Table 1: description: total set per site (not per stem), % fruit set per stem not per site
4. Table 3: without bold in the table
5. Fig. S1 & S3 - I propose to move to the main text
6. I suggest - if possible, add a small photo plateau with photos of species in habitat, flower, fruit, and seeds.
7. Editorial comments:
p.2 paragraph 2, l.7: [... 15, 16].
p.7. first paragraph 2x [13, 14].
p.7. section 3.5, second paragraph: [12, 14]
References: -please check carefully for some inaccuracies, e.g..
1. Estill, J. C., & .....
14. Madsen, D. L.
15. Pitts-Singer, T. L., Hanula, J. L ....
Reviewer 3 Report
The paper by Johnson et al. is a follow up to the earlier studies on the reproduction of a listed plant species from the flora of Florida, US. The study is well designed and clearly presented, however, it reports field results from only a single vegetation season, which in my opinion makes it highly speculative in the conclusions. For instance, the authors discuss the role of herbivory in the context of successful survival of the M. alba populations, but the observed increase in the herbivoiry pressure can be solely a result of some local phenomena that vanish the next year. Also seed production / seed set in plants is generally correlated with weather and habitat conditions and/or pollinator activity, and without any data on that any attempt to reveal the reasons for observed situation is more of a guessing. I would also advice to collect some more vegetation data that characterize the studied populations – for instance about the co-occuring (co-flowering) species that may contribute to the observed variation.